# Recurrence pattern predicts aneurysm rupture after coil embolization

**Iku Nambu**[ID], **Kouichi Misaki** *, **Takehiro Uno, Akifumi Yoshikawa, Naoyuki Uchiyama**[ID], **Masanao Mohri, Mitsutoshi Nakada**

Department of Neurosurgery, Kanazawa University School of Medicine, Ishikawa, Japan

* misaki@med.kanazawa-u.ac.jp

## Abstract

### Introduction

Hemorrhage from a recurrent aneurysm is a major concern after coiling for intracranial aneurysms. We aimed to identify aneurysm recurrence patterns associated with hemorrhage.

### Material and methods

We investigated radiological data of patients who underwent coiling for intracranial aneurysms in 2008–2016 and were followed-up for at least 6 months. Aneurysm recurrence patterns were classified as: type I, enlargement of aneurysm neck; type II, recurrent cavity within the coil mass; type III, recurrent cavity along the aneurysm wall; and type IV, formation of a daughter sac. We evaluated the incidence of various recurrence patterns with or without hemorrhage.

### Results

Of the 173 aneurysms included in the study (mean follow-up period, 32 months; range, 6–99 months), 22 (13%) recurred and required re-treatment. The recurrence patterns included type I, II, III, and IV in 7 (4%), 4 (2%), 9 (5%), and 2 (1%) cases, respectively. Most of the type I, II, and III recurrences occurred within 1 year, and type IV occurred at 7 years after coiling. Three aneurysms exhibited hemorrhage, one with type III and two with type IV pattern. The two aneurysms with type IV recurrence initially occurred as type I; however, the recurrent neck enlarged gradually, resulting in new sac formation.

### Conclusions

We recommend prompt re-treatment for aneurysms recurring with type III or IV patterns, as such patterns were associated with hemorrhage. Furthermore, we need a special care to type I recurrence with enlargement of recurrent neck because this specific pattern may develop to type IV.

**Data Availability Statement:** All relevant data are within the manuscript and its Supporting Information files.

**Funding:** The authors received no specific funding for this work.

**Competing interests:** The authors have declared
that no competing interests exist.

## Introduction

Endovascular embolization with detachable coils is widely performed and currently considered a safe, minimally invasive, and reliable technique to obliterate intracranial aneurysms [1–3]. However, aneurysm recurrence is more frequent after coil embolization than after surgical clipping [4, 5]. Several risk factors for recanalization after coil embolization have been proposed, including a ruptured aneurysm, large size, wide neck, posterior circulation location, and low volume embolization ratio (VER) [6–10]. Some studies using computational fluid dynamics analysis reported that hemodynamic forces may be related to the recanalization of coiled aneurysms [11, 12].

Hemorrhage from a recurrent aneurysm is associated with high mortality and morbidity [13–15]. To reduce the risks of hemorrhage after recurrence, angiographic follow-up and prompt re-treatment are necessary. The choice of re-treatment strategy depends on the aneurysm recurrence pattern. However, it remains unknown which recurrence pattern carries a higher risk of hemorrhage and would likely need re-treatment. The goal of the present study was to classify recurrent aneurysms by recurrence patterns and to assess the relationship of recurrence pattern with the need for re-treatment and the incidence of hemorrhage.

## Materials and methods

This study was reviewed and approved by the Independent Ethics Committee of Kanazawa University School of Medicine (No. 1781). Informed consent was obtained in writing from all patients or their next of kin.

### Patients and aneurysm data

We retrospectively investigated the clinical and radiological data on 231 aneurysms in 209 consecutive patients who underwent endovascular coiling for intracranial saccular aneurysms over a period of approximately 9 years (between January 2008 and June 2016). Only patients followed-up for more than 6 months after coiling were included. Fifty-eight aneurysms were excluded. The follow-up period of 55 aneurysms was shorter than 6 months, including 43 aneurysms followed up at another hospital and 12 aneurysms in patients who died within 6 months of endovascular coiling. One aneurysm that underwent a second coiling procedure at 3 months after the first procedure and two aneurysms with re-hemorrhage at 3 and 21 days after coiling were also excluded.

The final study population included 154 patients with 173 aneurysms. General patient information and aneurysm characteristics are summarized in Table 1. Posterior communicating artery (PcomA) and paraclinoid internal carotid artery aneurysms were the most frequent (19.7%), followed by anterior communicating artery (AcomA) aneurysms (15.0%).

### Endovascular procedure

Aneurysm coiling was performed with the patient under general anesthesia. Before coiling, a bolus of 3,000–5,000 IU of heparin was administered intravenously, followed by bolus infusion of 1,000 IU per hour. Anticoagulation aimed to maintain the activated clotting time at twice the normal value during catheterization and coil placement. Coiling was performed with various types of standard platinum coils. Stent or balloon assistance was indicated in case of wide-necked aneurysms. The initial angiographic results of coiling were graded according to the Modified Raymond-Roy Classification (MRRC) [16]: class I, complete obliteration; class II, residual neck; class IIIa, residual aneurysm with contrast within the coil interstices; class IIIb,

**Table 1. Characteristics of patients and aneurysms treated with endovascular coiling.**

| Characteristic | | Value (%) |
|---|---|---|
| Patients | | 154 |
| Aneurysms | | 173 |
| Age, years | | 58±11 |
| Female sex | | 111 (72) |
| Ruptured | | 64 (37) |
| Location | | |
| ICA | Cavernous | 1 (0.6) |
| | OphA | 7 (4.0) |
| | PcomA | 34 (19.7) |
| | Tip | 4 (2.3) |
| | Paraclinoid | 34 (19.7) |
| | Other | 14 (8.1) |
| ACA | Proximal | 3 (1.7) |
| | AcomA | 26 (15.0) |
| | Distal | 2 (1.2) |
| MCA | Proximal | 2 (1.2) |
| | Bifurcation | 10 (5.8) |
| | Distal | 3 (1.7) |
| PCA | P1 segment | 1 (0.6) |
| | Distal | 1 (0.6) |
| BA | Tip | 12 (6.9) |
| | Trunk | 4 (2.3) |
| | SCA | 5 (2.9) |
| VA | PICA | 7 (4.0) |
| | Other | 3 (1.7) |

ACA, anterior cerebral artery; AcomA, anterior communicating artery; BA, basilar artery; ICA, internal carotid artery; MCA, middle cerebral artery; OphA, ophthalmic artery; PCA, posterior cerebral artery; PcomA, posterior communicating artery; PICA, posterior inferior cerebellar artery; SCA, superior cerebellar artery; VA, vertebral artery

residual aneurysm with contrast along the aneurysm wall. VER was determined as the ratio of the coil volume to the aneurysm volume.

## Imaging follow-up and additional treatment

The patients were regularly followed up with MRI every 6 months for one year after treatment, and every year thereafter. In case of incompletely-treated aneurysms, follow-up MRI was taken more often. Digital subtraction angiography (DSA) was performed when recurrence was suspected on MRI. Major recurrence was defined as aneurysm recurrence needing re-treatment. Additional treatment was considered if: (1) contrast filling was noted in a cavity occupying > 20% of the volume of the original aneurysm in case of adequately-treated aneurysms or (2) contrast filling was further increased in case of incompletely-treated aneurysms or (3) hemorrhage was noted from the recurrent aneurysm. Minor recurrence was defined as aneurysm recurrence not needing re-treatment.

"Aneurysm growth" was defined as an enlargement of the aneurysm diameter on MRI or DSA. When coil mass became loose on skull anteroposterior view and lateral view, we called "coil compaction".

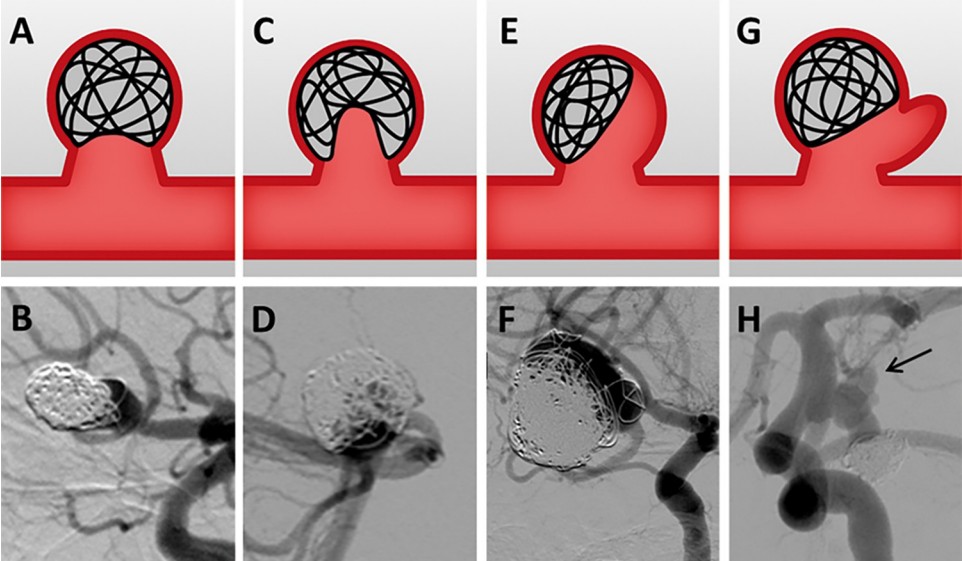

**Fig 1. Classification of aneurysm recurrence patterns.** (A and E) Type I, enlargement of aneurysm neck. (B and F) Type II, recurrent cavity within the coil mass. (C and G) Type III, recurrent cavity along the aneurysm wall. (D and H) Type IV, formation of a daughter sac. Black arrow indicates a daughter sac.

### Definition of recurrent patterns

We proposed the following classification of aneurysm recurrence patterns: type I, enlargement of the aneurysmal neck due to coil compaction or neck growth (Fig 1A and 1E); type II, recurrent cavity within the coil mass (Fig 1B and 1F); type III, recurrent cavity along the aneurysm wall (Fig 1C and 1G); and type IV, formation of a daughter sac (Fig 1D and 1H). We stratified re-treated aneurysms according to recurrence pattern and subsequently identified the recurrence patterns associated with hemorrhage and the need for re-treatment.

### Statistical analysis

Univariate analysis was performed for intergroup comparisons, as appropriate. The data are reported as means ± standard deviation for continuous variables and the number of observations (frequency, %) for categorical variables. The statistical significance was analyzed using the Fisher's exact test for categorical, nominal variables, or the Mann-Whitney U test for continuous, numerical variables. Multivariate logistic regression analysis included all variables that were found to be significant on univariate analysis at a P-value of $< 0.05$. Results of logistic regression were reported as odds ratio (OR) with P-value $< 0.05$ for a 95% confidence interval (CI), which was considered statistically significant. Univariate and multivariate analyses were performed using SPSS (IBM SPSS Statistics 23, Chicago, IL, USA).

Kruskal-Wallis one-way non-parametric ANOVA was performed for each parameter (maximum diameter, neck width, and volume embolization ratio) for each of the five groups (no re-treatment, recurrence type I, type II, type III, and type IV). Post-hoc pairwise group comparisons were performed using Dunnett's multiple-testing correction. Each re-treatment group (recurrence type I, II, III, and IV) was compared with the control group of no re-treatment group. Statistical analyses were performed with EZR (Saitama Medical Center, Jichi Medical University, Saitama, Japan), which is a graphical user interface for R (The R Foundation for Statistical Computing, Vienna, Austria). A value of α = 0.05 was selected as the significance threshold.

## Results

### Factors associated with re-treatment after coil embolization

Of the 173 aneurysms included in this study, 22 had major recurrence required re-treatment (12.7%). Factors associated with re-treatment after coil embolization are summarized in Table 2. In the re-treatment group, the proportion of patients with ruptured aneurysms was significantly higher (P<0.001), maximum size and neck width were significantly larger (P = 0.003, 0.002), the proportion of BA and MRRC class IIIb were significantly higher (P = 0.032, 0.025), VER was significantly lower (P<0.001) than in the no re-treatment group. No significant difference was found in terms of using balloon or stent. In multivariate analysis, rupture status (OR 18.20, 95% CI 4.00–82.60, P<0.001) and maximum diameter (OR 1.25, 95% CI 1.00–1.56, P = 0.049) were statistically significant.

### Recurrence patterns associated with re-treatment

Of the 173 coiled aneurysms in this study, 151 had no re-treatment, and 22 had major recurrence requiring re-treatment (12.7%), showing the following recurrence patterns; type I, 7 aneurysms (4.0%); type II, 4 aneurysms (2.3%); type III, 9 aneurysms (5.2%); and type IV, 2 aneurysms (1.2%). The aneurysm characteristics and results of coiling are summarized in Table 3.

In the Type I, four of 7 patients of Type I had major recurrence 6 months after coiling. Three of 7 patients of Type I had minor recurrence 6 months after coiling, and developed

**Table 2. Factors associated with re-treatment after coil embolization.**

| | | Univariate analysis | | | Multivariate analysis | | |
|---|---|---|---|---|---|---|---|
| | | No re-treatment n = 151 | Re-treatment n = 22 | P | OR | 95% CI | P |
| Age | | 58.1±10.9 | 60.4±11.8 | 0.283 | | | |
| Female (%) | | 109 (72.2) | 13 (59.1) | 0.218 | | | |
| Rupture (%) | | 46 (30.5) | 18 (81.8) | <0.001 | 18.20 | 4.00–82.60 | <0.001 |
| Location | | | | | | | |
| ICA (%) | | 86 (57.0) | 8 (36.3) | 0.107 | | | |
| ACA (%) | | 25 (16.6) | 6 (27.3) | 0.237 | | | |
| MCA (%) | | 13 (8.6) | 2 (9.1) | 1.000 | | | |
| PCA (%) | | 2 (1.3) | 0 | 1.000 | | | |
| BA (%) | | 15 (9.9) | 6 (27.3) | 0.032 | 1.40 | 0.26–7.53 | 0.694 |
| VA (%) | | 10 (6.6) | 0 | 0.365 | | | |
| Size | | | | | | | |
| maximum, mm | | 7.8±3.2 | 10.2±4.2 | 0.003 | 1.25 | 1.00–1.56 | 0.049 |
| neck width, mm | | 4.0±1.9 | 5.6±2.3 | 0.002 | 0.95 | 0.65–1.40 | 0.802 |
| Coiling procedure | | | | | | | |
| balloon-assisted (%) | | 36 (23.8) | 4 (18.1) | 0.787 | | | |
| stent-assisted (%) | | 15 (9.9) | 2 (9.1) | 1.000 | | | |
| Result | | | | | | | |
| MRRC Class I (%) | | 46 (30.5) | 3 (13.6) | 0.131 | | | |
| MRRC Class II (%) | | 70 (46.4) | 8 (36.4) | 0.493 | | | |
| MRRC Class IIIa (%) | | 29 (19.2) | 7 (31.8) | 0.172 | | | |
| MRRC Class IIIb (%) | | 6 (3.9) | 4 (18.2) | 0.025 | 5.37 | 0.60–48.20 | 0.133 |
| VER, % | | 24.2±5.3 | 19.8±5.3 | <0.001 | 0.96 | 0.86–1.07 | 0.432 |

MRRC, Modified Raymond-Roy Classification; VER, volume embolization ratio

**Table 3. Characteristics of recurrence patterns.**

| | No re-treatment n = 151 (87.3%) | Recurrence pattern | | | | P |
|---|---|---|---|---|---|---|
| | | Type I n = 7 (4.0%) | Type II n = 4 (2.3%) | Type III n = 9 (5.2%) | Type IV n = 2 (1.2%) | |
| Initially ruptured (%) | 46 (31) | 6 (86) | 3 (75) | 7 (78) | 2 (100) | |
| Size | | | | | | |
| maximum diameter, mm | 7.8±3.2 | 9.4±6.5 | 10.7±1.2 | 11.1±3.0* | 7.3±1.6 | 0.0045 |
| neck width, mm | 4.0±1.9 | 3.8±2.0 | 7.6±1.2* | 6.5±1.7* | 3.4±1.6 | 0.0002 |
| Coiling procedure | | | | | | |
| balloon assisted (%) | 36 (23.8) | 2 (29) | 1 (25) | 1 (11) | 0 | |
| stent-assisted (%) | 15 (9.9) | 0 | 1 (25) | 1 (11) | 0 | |
| Results | | | | | | |
| MRRC class I (%) | 46 (31) | 2 (29) | 0 | 1 (11) | 0 | |
| MRRC class II (%) | 70 (46) | 2 (29) | 3 (75) | 2 (22) | 1 (50) | |
| MRRC class IIIa (%) | 29 (19) | 3 (43) | 1 (25) | 2 (22) | 1 (50) | |
| MRRC class IIIb (%) | 6 (4) | 0 | 0 | 4 (44) | 0 | |
| VER, % | 24.2±5.3 | 22.6±4.7 | 18.8±1.4* | 17.2±6.2* | 23.5±2.1 | 0.0053 |
| Major recurrence | | | | | | |
| within 6 months (%) | | 4 (57) | 2 (50) | 8 (89) | 0 | |
| from 6 to 12 months (%) | | 3 (43) | 1 (25) | 1 (11) | 0 | |
| from 12 to 18 months (%) | | 0 | 1 (25) | 0 | 0 | |
| Aneurysm growth (%) | | 1 (14) | 0 | 0 | 1 (50) | |
| Coil compaction (%) | | 4 (57) | 0 | 1 (11) | 0 | |
| Aneurysm growth + Coil compaction (%) | | 2 (29) | 4 (100) | 8 (89) | 1 (50) | |
| Recurrent hemorrhage (%) | | 0 | 0 | 1 (11) | 2 (100) | |

Kruskal-Wallis tests

*Variable significantly compared to no re-treatment group

MRRC, Modified Raymond-Roy Classification; VER, volume embolization ratio

major recurrence 12 month after coiling. The proportion of coil compaction is higher than other recurrence patterns. In the Type II, the neck width was significantly larger (P = 0.007), and VER was significantly lower (P = 0.038) than in the no re-treatment group. Two of 4 patients of Type II had major recurrence 6 months after coiling. Another 2 patients of Type II had major recurrence 12 and 18 months after coiling, respectively. Aneurysm growth and coil compaction were noted in all of Type II recurrence aneurysms. In the Type III, the maximum diameter and neck width were significantly larger (P = 0.005, 0.001) than in the no re-treatment group. The proportion of patients with MRRC class IIIb was higher than any other group. VER was significantly lower (P = 0.028) than in the no re-treatment group. Eight of 9 patients of Type III had major recurrence within 6 months after coiling. Aneurysm growth and coil compaction were noted in 8 patients. No type IV recurrence occurred within 18 months. The time from coiling to recurrence of the two type IV aneurysms was 83 and 94 months.

## Recurrence patterns associated with hemorrhage

Hemorrhage from recurrent aneurysms occurred in three patients, corresponding to an overall incidence of 1.7% (3/173). Relevant clinical and imaging data of the three patients with re-hemorrhage are summarized in Table 4.

**Table 4. Clinical and imaging data of the three patients with hemorrhage from recurrent aneurysm.**

| Case | Age | Sex | Location | Size, mm | H&K grade | WFNS grade | VER | MRRC class | Time from coiling to hemorrhage | Recurrence pattern | Re-treatment |
|---|---|---|---|---|---|---|---|---|---|---|---|
| 1 | 59 | F | BA tip | 12.0 | 3 | 2 | 10.4 | IIIb | 6 months | Type III | Coiling |
| 2 | 63 | F | ICA-PcomA | 8.4 | 1 | 1 | 24.9 | II | 83 months | Type IV | Coiling |
| 3 | 59 | F | BA tip | 6.1 | 2 | 1 | 21.5 | IIIa | 94 months | Type IV | Coiling |

BA, basilar artery; F, female; H&K, Hunt-Kosnik; ICA, internal carotid artery; MRRC, Modified Raymond-Roy Classification; PcomA, posterior communicating artery; VER, volume embolization ratio; WFNS, World Federation of Neurological Surgeon

The basilar tip aneurysm in case #1 (Fig 2A) was embolized, resulting in incomplete occlusion with a VER of 10.4% (Fig 2B). No change was observed on MRA 1 month after coiling, but recurrence with type III was observed on MRA 4 months after coiling. Re-hemorrhage occurred 2 months after final MRA (Fig 2C), and the patient underwent re-coiling.

In cases #2 and #3, re-hemorrhage occurred extremely late after coiling, with type IV recurrence in both patients. The PcomA aneurysm in case #2 (Fig 3A) was embolized with coils. Angiography indicated that initial embolization resulted in an occlusion of MRRC class II with a VER of 24.9% (Fig 3B). Follow-up annual MRA showed that the remnant neck had gradually increased in size until 6th year (Fig 3C and 3D), eventually giving rise to a new sac at 7th year after coiling. Although re-treatment was scheduled, re-hemorrhage occurred 2 months after the final MRA (Fig 3E). The basilar tip aneurysm in case #3 recurred at 6 years after first coiling (Fig 4A). A second coiling procedure was performed, resulting in occlusion of MRRC class IIIa with a VER of 21.5% (Fig 4B). This aneurysm recurred with neck enlargement (Fig 4C and 4D), eventually, a new sac was formed at 7th year after treatment. Although re-treatment was scheduled, re-hemorrhage occurred 2 months after the final MRA (Fig 4E).

## Discussion

In the present study, we evaluated risk factors associated with re-treatment after coil embolization. Moreover, we proposed a new classification of recurrence patterns and detected recurrence patterns associated with re-treatment and hemorrhage.

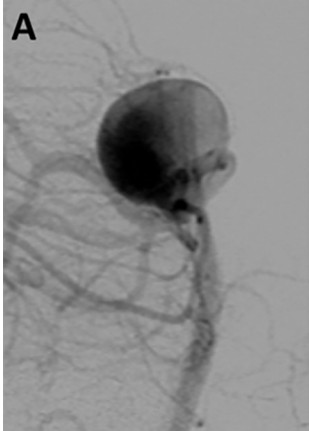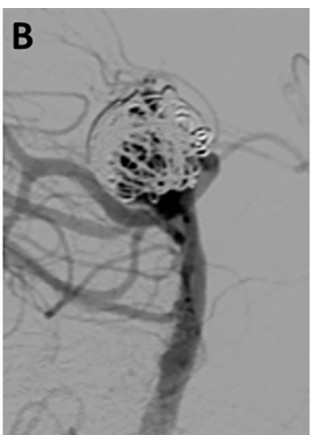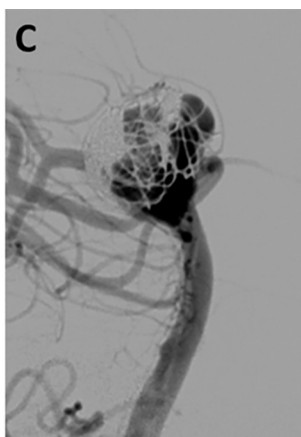

**Fig 2. Recurrent hemorrhage after coil embolization of an intracranial aneurysm in a 59-year-old woman (case #1).** (A and B) Basilar tip aneurysm embolized using coils. Initial embolization was angiographically graded as MRRC class IIIb, with a VER of 10.4%. (C) Re-hemorrhage occurred at 6 months after coiling, with angiography indicating type III recurrence (i.e., recurrent cavity along the aneurysm wall).

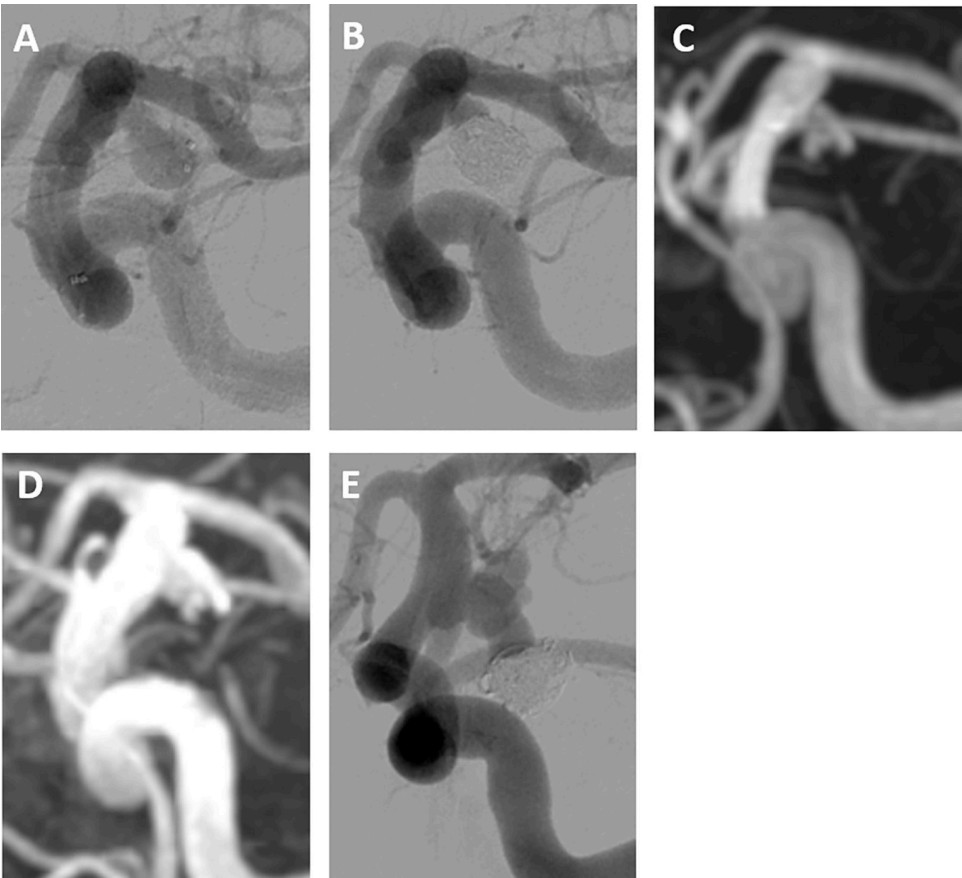

**Fig 3. Recurrent hemorrhage after coil embolization of an intracranial aneurysm in a 64-year-old woman (case #2).** (A and B) Posterior communicating artery aneurysm embolized using coils. Initial embolization was angiographically graded as MRRC class II, with a VER of 24.9%. (C) MRA at 2 years after coiling indicated a remnant neck. (D) MRA at 5 years after coiling indicated enlargement of the recurrent neck. (E) Re-hemorrhage at 7 years after coiling, with angiography indicating the formation of a daughter sac.

In the International Subarachnoid Aneurysm Trial (ISAT) [3], late re-treatments (later than 3 months after the coiling) were performed in 9.0% (94/1,045) of patients after aneurysm recurrence or rebleeding. The mean interval to retreatment was 20.7 months (range, 3 to 80 months). Younger age, larger lumen size, and incomplete occlusion were risk factors for late re-treatment after coiling. In this study, re-treatments were performed in 12.7% (22/173). Ruptured aneurysms, larger size, wider neck, BA location, MRRC class IIIb, and lower VER correlated with re-treatment. In particular, rupture status and maximum diameter were risk factors for re-treatment after coiling.

The Modified Raymond-Roy Classification [16] used for the angiographic results after coil embolization is useful to predict recurrence or progressive occlusion, but it is difficult to express with or without recurrence. Therefore, in this study, we proposed a novel classification limited to recurrence aneurysm after coil embolization. We divided recurrence aneurysms into 4 types of recurrence patterns, and detected recurrence patterns associated with re-treatment and hemorrhage. Larger size and wider neck ruptured aneurysms with low VER tended to recur with Type II or III. In the Type III recurrence group, the proportion of MRRC class IIIb aneurysms was higher than other recurrence types. Almost of all Type III recurrence occurred within 6 months. Recurrent hemorrhage was associated with Type III and IV recurrence.

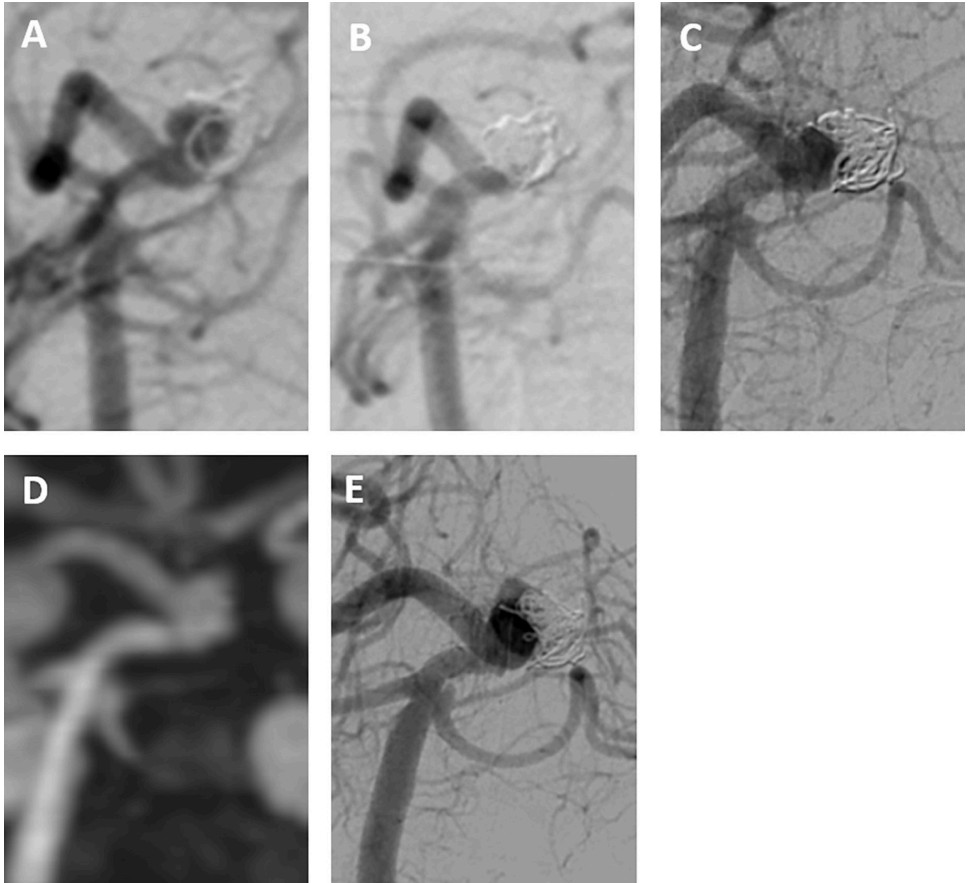

**Fig 4. Recurrent hemorrhage after coil embolization of an intracranial aneurysm in a 59-year-old woman (case #3).** (A and B) Basilar tip aneurysm recurring at 6 years after first coiling, treated via a second coiling procedure. Embolization via second coiling was evaluated angiographically as MRRC class IIIa, with a VER of 21.5%. (C) Digital subtraction angiography at 5 years after second coiling indicating a recurrent neck. (D) MRA at 6 years after second coiling, indicating enlargement of the recurrent neck. (E) Hemorrhage at 7 years after coiling, with angiography indicating the formation of a daughter sac.

Hemorrhage from a recurrent aneurysm is typically classified as early or late hemorrhage, with early hemorrhage defined as re-bleeding within 30 days after coiling associated with worsening of the patient's condition. In our study, we excluded two patients with early re-hemorrhage. Thrombosis is formed in true aneurysms or pseudoaneurysms after rupture, and subsequent thrombus resolution after coiling can lead to reopening of the aneurysm lumen, resulting in early re-hemorrhage [13]. These recurrence mechanisms underlying early re-hemorrhage are completely different from those associated with late hemorrhage.

Late hemorrhage from a recurrent aneurysm is defined as the occurrence of re-bleeding later than 1 month after coiling [17, 18]. In our study, the incidence of the late re-bleeding was 1.7% (3/173), with type III and IV recurrence patterns. The patient described as case #1 had re-hemorrhage with type III recurrence at 6 months after coiling. The cause of re-hemorrhage appeared to be a low VER. Sluzewski et al. [19] described five cases of late re-hemorrhage of ruptured aneurysms treated with coils. Risk factors for late re-hemorrhage include large aneurysm size and incomplete aneurysm occlusion after initial embolization or on follow-up. Angiograms after re-hemorrhage were available in three of the described cases, which had occurred at 12, 30, and 40 months after initial coiling. In all three cases, the recurrence pattern

was of type III. Two case reports of delayed rupture of a previously coiled unruptured aneurysm indicated that aneurysms in the middle cerebral artery bifurcation and AcomA ruptured at 18 and 23 months, respectively, after initial coiling [20, 21]. Both these aneurysms showed recanalization with a type III pattern.

The aneurysms in cases #2 and #3 recurred with type IV patterns extremely late phase after coiling. In both cases, the aneurysm was originally of type I but progressed to type IV, which is in agreement with observations from longitudinal follow-up. Liu et al. [22] reported delayed rupture of a previously coiled unruptured AcomA aneurysm. This aneurysm recurred with a type I pattern at 8 months after coiling, and a new sac was formed from the recurrent neck (type IV pattern) at 38 months after coiling. Recurrent hemorrhage from the new sac occurred one day after final angiography. We need a special care to type I recurrence with enlargement of recurrent neck because this specific pattern may develop to type IV.

## Conclusions

We recommend immediate re-treatment in patients with recurrent aneurysms of type III or IV because these recurrence patterns were associated with hemorrhage. While no hemorrhage was noted for aneurysms with type I recurrence, new sac is sometimes formed from the recurrent neck after a long time follow-up, resulting in hemorrhage. Therefore, it may be advisable to follow carefully for type I recurrence.

## Supporting information

**S1 Dataset.**
(XLSX)

## Acknowledgments

We thank Editage (https://www.editage.jp) for English language review of the manuscript.

## Author Contributions

**Conceptualization:** Iku Nambu, Takehiro Uno, Akifumi Yoshikawa, Naoyuki Uchiyama, Masanao Mohri, Mitsutoshi Nakada.

**Data curation:** Iku Nambu.

**Formal analysis:** Iku Nambu.

**Investigation:** Iku Nambu.

**Methodology:** Iku Nambu.

**Project administration:** Iku Nambu.

**Supervision:** Kouichi Misaki.

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
