## [Decision Letter · Decision Letter 0]

31 Dec 2020

PONE-D-20-27179

Recurrence pattern predicts aneurysm rupture after coil embolization

PLOS ONE

Dear Dr. Misaki,

Thank you for submitting your manuscript to PLOS ONE. After careful consideration, we feel that it has merit but does not fully meet PLOS ONE’s publication criteria as it currently stands. Therefore, we invite you to submit a revised version of the manuscript that addresses the points raised during the review process.

Please revise by addressing all reviewers' comments. In particular, provide new statistical analysis (multi-variable analysis for risk factors regarding recurrence and aneurysmal rupture. Also provide more CFD data (as outlines by reviewer 3).

We look forward to receiving your revised manuscript.

Kind regards,

Stephan Meckel, MD, PhD

Academic Editor

PLOS ONE

Journal Requirements:

Reviewers' comments:

Reviewer's Responses to Questions

**Comments to the Author**

1. Is the manuscript technically sound, and do the data support the conclusions?

Reviewer #1: No

Reviewer #2: Yes

Reviewer #3: Partly

2. Has the statistical analysis been performed appropriately and rigorously? 

Reviewer #1: No

Reviewer #2: Yes

Reviewer #3: Yes

3. Have the authors made all data underlying the findings in their manuscript fully available?

Reviewer #1: Yes

Reviewer #2: Yes

Reviewer #3: Yes

4. Is the manuscript presented in an intelligible fashion and written in standard English?

Reviewer #1: Yes

Reviewer #2: Yes

Reviewer #3: Yes

5. Review Comments to the Author

Reviewer #1: This is an interesting paper about the recurrence pattern prediction of aneurysm rupture after coil embolization. However, aneurysm rupture occurred only in three cases. Therefore, the significance of these findings is very low.

Reviewer #2: Authors should specify the initial treatment of the recurrent aneurysms (simple coling/ballon or stent assisted coiling etc.) in order to evaluate whether does a relation exist with the recurrence. It would be of some interest tie the recurrence type with the initial technique used to coil the aneurysm.

Reviewer #3: The authors present a single institution retrospective review of coiled cerebral aneurysms with a focus on recurrences, and they propose a grading system to characterize recurrence patterns. While this topic is interesting, I believe that they authors should revise their analysis of this population to make a more meaningful conclusion.

1) Please state very clearly how recurrence was defined in incompletely-treated aneurysms.

2) It is nice that you focus on the three cases that hemorrhaged as this is ultimately the event that we would like to avoid in coiled aneurysm patients. Please report as much as possible regarding the results of interval scans in these patients so that we can better understand the natural history of aneurysm recurrence.

3) Along similar lines, reporting the results of interval scans for all of the recurrent aneurysms would be valuable. The occurrence and rate of aneurysm growth and/or coil compaction may be just as important a finding as the pattern of recurrence.

4) Although risk factors for aneurysm recurrence after coiling have been previously described, this manuscript would be much stronger if you performed a multivariate analysis of risk factors associated with recurrence (both overall and for each recurrence pattern) and hemorrhage.

5) Please discuss whether you think there is a continuum between your recurrence patterns.

6) The CFD data is interesting but adds little to this study since it was only used for two examples. If possible, this manuscript would be significantly stronger with a more complete CFD analysis of recurrent aneurysms.

7) Please include percentages of patients in each recurrence group (rather than just absolute numbers) in the Abstract, Results, and Tables

6. PLOS authors have the option to publish the peer review history of their article (what does this mean?). If published, this will include your full peer review and any attached files.

Reviewer #1: No

Reviewer #2: No

Reviewer #3: No

---

## [Author Response · Author response to Decision Letter 0]

22 Feb 2021

February 14, 2021

Stephan Meckel, MD, PhD

Academic Editor

PLOS ONE

Dear Editor: 

I would like to re-submit an original article for publication in PLOS ONE, titled “Recurrence pattern predicts aneurysm rupture after coil embolization.” The manuscript number is PONE-D-20-27179.

The manuscript has been carefully rechecked and appropriate changes have been made in accordance with the reviewers’ suggestions. The responses to their comments have been prepared and attached herewith.

We thank you and the reviewers for your thoughtful suggestions and insights, which have enriched the manuscript and produced a more balanced and better account of the research. We hope that the revised manuscript is now suitable for publication in your journal.

I look forward to your reply.

Sincerely,

Kouichi Misaki

Department of Neurosurgery, Graduate School of Medical Science, Kanazawa University

13-1 Takara-machi, Kanazawa

Ishikawa 920-8641, Japan

Phone: +81-76-265-2384

Fax: +81-76-234-4262

Email: misaki@med.kanazawa-u.ac.jp

---

## [Decision Letter · Decision Letter 1]

21 Jun 2021

PONE-D-20-27179R1

Recurrence pattern predicts aneurysm rupture after coil embolization

PLOS ONE

Dear Dr. Misaki,

Thank you for submitting your manuscript to PLOS ONE. After careful consideration, we feel that it has merit but does not fully meet PLOS ONE’s publication criteria as it currently stands. Therefore, we invite you to submit a revised version of the manuscript that addresses the points raised during the review process.

Please address all my comments below!

We look forward to receiving your revised manuscript.

Kind regards,

Stephan Meckel, MD, PhD

Academic Editor

PLOS ONE

Journal Requirements:

Additional Editor Comments (if provided):

Special comments from Editor:

- please leave the CFD analysis out completely as it has no relevance with only addressing 5 cases!

- No statistical correction for multiple testing of your dataset has been performed - please contact dedicated statistician to solve this problem - your dataset was tested with many statistical tests for many different questions!

Reviewers' comments:

Reviewer's Responses to Questions

**Comments to the Author**

1. If the authors have adequately addressed your comments raised in a previous round of review and you feel that this manuscript is now acceptable for publication, you may indicate that here to bypass the “Comments to the Author” section, enter your conflict of interest statement in the “Confidential to Editor” section, and submit your "Accept" recommendation.

Reviewer #1: All comments have been addressed

2. Is the manuscript technically sound, and do the data support the conclusions?

Reviewer #1: Yes

3. Has the statistical analysis been performed appropriately and rigorously? 

Reviewer #1: Yes

4. Have the authors made all data underlying the findings in their manuscript fully available?

Reviewer #1: Yes

5. Is the manuscript presented in an intelligible fashion and written in standard English?

Reviewer #1: Yes

6. Review Comments to the Author

Reviewer #1: I have no objections. The authors have adequately addressed all comments raised in a previous round of review.

7. PLOS authors have the option to publish the peer review history of their article (what does this mean?). If published, this will include your full peer review and any attached files.

Reviewer #1: No

---

## [Author Response · Author response to Decision Letter 1]

2 Aug 2021

We thank for your thoughtful suggestions and insights, which have enriched the manuscript and produced a more balanced and better account of the research. We hope that the revised manuscript is now suitable for publication in your journal.

---

## [Editor Report · Decision Letter 2]

23 Aug 2021

PONE-D-20-27179R2

Recurrence pattern predicts aneurysm rupture after coil embolization

PLOS ONE

Dear Dr. Misaki,

Thank you for submitting your manuscript to PLOS ONE. After careful consideration, we feel that it has merit but does not fully meet PLOS ONE’s publication criteria as it currently stands. Therefore, we invite you to submit a revised version of the manuscript that addresses the points raised during the review process.

Please find my additional comments below at the bottom of this letter.

We look forward to receiving your revised manuscript.

Kind regards,

Stephan Meckel, MD, PhD

Academic Editor

PLOS ONE

Additional Editor Comments (if provided):

1. Thank you for the added statistics with regards to multiple testing correction - any changes to the results?

2. As stated previously, CFD should be left out from the paper: I still see in in the methods section as well as results & discussion - please correct!
---

## [Author Response · Author response to Decision Letter 2]

6 Oct 2021

Comment 1: “Thank you for the added statistics with regards to multiple testing correction – any changes to the results?”

Response: Thank you for taking the time to read our manuscript and reply. In the previous results, the maximum diameter and neck width (Type II and III) were significantly larger, and VER (Type II and III) were significantly lower than in the no re-treatment group. In the multiple testing correction, there was no statistically significant difference in the maximum diameter in the Type II.

Comment 2: “CFD should be left out from the paper.”

Response: According to your suggestion, we have left out the CFD analysis from Abstract, Methods, Results, and Discussion. Instead of CFD analysis, we have added some words (page 2; line 35-36, page2; line 38-40, page 23; line 347-348, page 23; line 353-355).

---

## [Editor Report · Decision Letter 3]

16 Dec 2021

Recurrence pattern predicts aneurysm rupture after coil embolization

PONE-D-20-27179R3

Dear Dr. Misaki,

We’re pleased to inform you that your manuscript has been judged scientifically suitable for publication and will be formally accepted for publication once it meets all outstanding technical requirements.

Kind regards,

Stephan Meckel, MD, PhD

Academic Editor

PLOS ONE
---

## [Editor Report · Acceptance letter]

1 Sep 2022

PONE-D-20-27179R3 

Recurrence pattern predicts aneurysm rupture after coil embolization 

Dear Dr. Misaki:

I'm pleased to inform you that your manuscript has been deemed suitable for publication in PLOS ONE. Congratulations! Your manuscript is now with our production department. 

Kind regards, 

on behalf of

Prof. Dr. Stephan Meckel 

Academic Editor

PLOS ONE